# MRBicopter： Modular Reconfigurable Transverse Tilt-rotor Bicopter System

1st Qianyao Pan
School of Automation
Engineering
University of Electronic Science
and Technology of China
Chendu, China
panqianyaoupc@163.com

2nd Xin Lu
School of Automation
Engineering
University of Electronic Science
and Technology of China
Chendu, China
luxin_uestc@163.com

3rd Weijun Yuan
School of Automation
Engineering
University of Electronic Science
and Technology of China
Chendu, China
ywj861087955@163.com

4th Fusheng Li*
School of Automation
Engineering
University of Electronic Science
and Technology of China
Chendu, China
lifusheng@uestc.edu.cn

*Abstract*—This paper introduces a modular UAV(MRBicopter) that can realize structural combination reconstruction. Each module contains a rotor tilting structure and an active docking mechanism. By separating and combining submodules, the UAV functions can match the requirements of different flight tasks in real time. First, we designed the mechanical actuator to allow physically connected assembly to fly collaboratively. Secondly, according to different reconstructed structures, we propose two generalized control strategies to realize the independent control of posture through the reassignment of rotor speed and tilt angle.The feasibility of the mechanical design and control method is verified by simulation and ground experiment under ambient wind interference .

*Keywords—Reconfigurable and modular robots, bicopter, active docking mechanism, rotor tilting, wind interference, simulation.*

## I. INTRODUCTION

In recent years, multi-rotor UAVs have received a lot of attention due to their simplicity, agility and versatility. Research in multi-rotor UAVs has extended to air maneuvering, collective behavior, multi-modal motion, and modular reconfigurable robots[1]-[5]. Among them, the advantages brought by the modular and reconfigurable capabilities of UAVs are increasingly reflected. For example, in the context of disaster relief, modular reconfigurable robots can realize adaptability to different task scenarios through structural reconstruction, such as cooperating in the transportation of large items[16] and completing search and rescue tracking in complex environments[17].

In order to improve the stability and safety of modular reconfigurable UAV. Reference[6] designed an airborne self-assembled flying robot, ModQuad, which is composed of flexible flight modules and can easily move in a three-dimensional environment. For airborne real-time separation systems, a new deformable multi-link aerial robot is proposed in reference[7], which consists of a link module of a 1-DOF thrust vector mechanism, and a transformation planning method is proposed to ensure the minimum force/moment by taking into account the 1-DOF thrust vector angle. Design for separation structure; Reference[8] proposes a magnetic-based connection mechanism, which uses a lightweight passive mechanism to dock and unload in mid-air. Aiming at the application scenario of modular UAV, a self-assembly robot based on autonomous

module was proposed in literature[9], which can fly together and assemble into rectangular structures in the air. Literature[10] proposes a full-attitude geometric control algorithm for synchronous tilting hexagonal rotorcraft to realize arbitrary Angle flight at the cost of efficiency. In literature[11], a tilt-rotor UAV was designed. The tilt-rotor mechanism can restrain power dissipation and has a wider inclination range. In literature[12], a structure connecting two helicopter modules is designed, which can fly along any Angle of the wall; Literature[13] proposed the idea of splitting quadcopter UAV into two twin-rotor UAV in real time in the air and developed the modular quadcopter(SplitFlyer). Literature[14] developed a combinable and extensible tilt-rotor UAV(CEDTR), which can match different task scenarios by changing the combination and number of unmanned sub-modules. Literature[15] developed an airborne detachable quadrotor UAV suitable for narrow gaps, which improved the environmental adaptability of reconfigurable UAV.

In this paper, we design a transverse two-rotor tilting bicopter that can be combined and reconstructe, called modular reconfigurable bicopter(MRBicopter),which can not only realize cooperative flight in single module state, but also can get multi-module combination flight control.The main contributions of this paper are in three aspects:

1) Modular reconfigurable bicopter with rotor vector tilting structure and active combination docking mechanism is designed and modeled, which can realize structural reconfiguration to adapt to different task requirements.

2) The UAV dynamics model is built and the UAV control distribution and controller design are completed to realize the control of a single module and the full degree of freedom control of the assembly.

3) The environmental wind interference module is innovatively introduced in the simulation to make the simulation result more close to the reality.

The structure of this paper is as follows: Section II introduces the structure of MRBicopter. Section III describes the modeling of MRBicopter. Section IV shows the control distribution and controller design of MRBicopter. Section V demonstrates the results of simulation and tests. The conclusions are presented in section VI.

---

*The author is the corresponding author.

XXX-X-XXXX-XXXX-X/XX/$XX.00 ©20XX IEEE

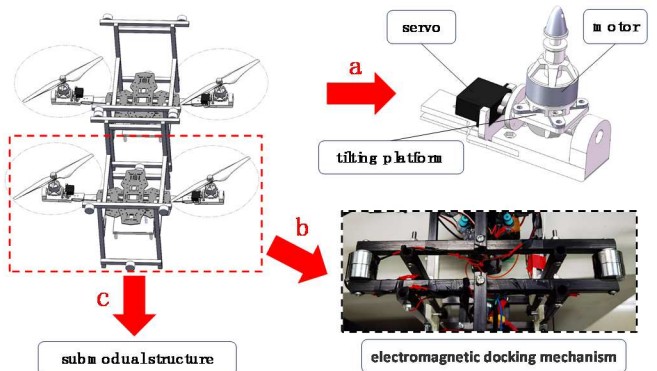

Fig.1: MRBicopter mechanical structure. (a) rotor vector tilting structure, (b) electromagnet combination butt mechanism, (c) submodule structure.

## II. DESIGN

### A. Rotor vector tilting structure

The rotor propeller axis of the traditional UAV is fixed, which direction of lift force cannot be changed. Here, we adopt the design of rotor vector tilting structure(Fig.1(a)). The rotor can tilt around the arm shaft, and each rotor is separately installed with a servo steering machine to control the tilting angle. This structure increases the input of UAV assembly control quantity, and can realize the full freedom control of MRBicopter assembly.

### B. Electromagnet combination butt structure

For the docking device between modular reconfigurable MRBicopter, permanent magnets(NdFe) are used in traditional reconfigurable UAVs. This scheme has a slow control response during separation and is easy to cause instability. Therefore, we designed a multi-locking electromagnet combination docking mechanism(Fig.1(b)).It uses a circular electromagnet as the main actuator, and realizes the on-off of the electromagnet by using a program to control the relay. A total of three locking nodes are included, each node can provide 5KG of locking suction.

### C. Electromagnet combination butt structure

MRBicopter consists of two cross-mounted bicopter single modules(Fig.1(c)). The single module can not only realize autonomous cooperative flight, but also complete assembly reconstruction by magnetic attraction.

## III. DYNAMICS

### A. Establishment of the frame

In this section, four different frames are introduced to define the flight attitude of MRBicopter(Fig.2). The rotation frame system as follows.

1) World frame $W_E$ .World frame is fixed coordinate system.

2) Assembly frame $B_z$ .The origin of the $B_z$ is located at the center of mass of the assembly, and its position relative to the world frame is expressed as $P_W = \{ x_w \quad y_w \quad z_w \}$ ; Relative

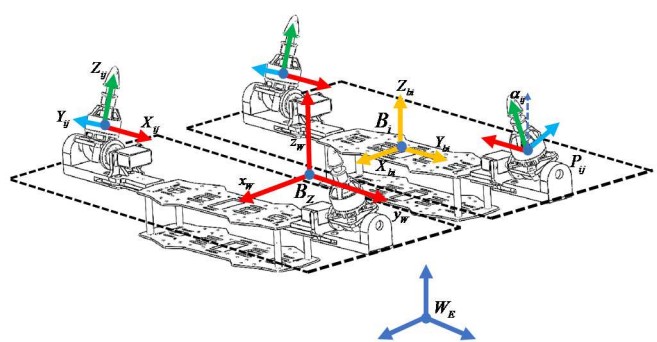

Fig.2: MRBicopter frame system Settings.

velocity is expressed as $V_W = \{ V_{WX} \quad V_{WY} \quad V_{WZ} \}$ ;The angular velocity of the assembly is expressed as $\Omega = \begin{bmatrix} \omega_x & \omega_y & \omega_z \end{bmatrix}^T$ ; The attitude angle is expressed as $\Theta = \{ \phi \quad \theta \quad \psi \}^T$ , where $\phi$ is the roll angle, $\theta$ is the pitch angle, and $\psi$ is the yaw angle .

3) Submodule frame $B_i$ . The origin of the submodule frame is located at the centroid of the submodule MRBicopter, which is defined as $\{ X_{bi} \quad Y_{bi} \quad Z_{bi} \}$ . The Euler angle in the submodule frame $B_i$ is expressed as $\Theta_i = \begin{bmatrix} \phi_i & \theta_i & \psi_i \end{bmatrix}^T$ .

4) Rotor frame $P_{ij}$ .The origin of the rotor frame is located at the position of the rotor motor centroid, the z axis points to the rotor lift direction, and the x axis points to the body centroid. The tilt angle of the rotor is set as $\alpha_{ij}$ .

### B. Derivation of Dynamics and Kinematic Model

In this section, we will deduce the attitude dynamics and kinematics equations of MRBicopter, which will eventually be used in the control allocation and controller model construction in section 4. The i-th submodule in the assembly has two rotors, which are distributed on an axis. The rotor speed is expressed as $\varpi_{ij}$ . Therefore, the lift force and rotation torque generated by the j-th rotor in the module can be written as:

$$f_{ij} = K_T \varpi_{ij}^{\ 2} \qquad (1)$$

$$\tau_{ij} = K_Q \varpi_{ij}^{\ 2} \qquad (2)$$

Where, $K_T, K_Q$ is the rotor motor constant.

In the Assembly frame $B_z$ , MRBicopter's lift force $F_B$ is as follows.

$$F_{ij}^B = f_{ij}^{\ \{B_z\}} R_{[P_{ij}]}(\alpha_{ij}) E$$
$$F_B = \sum_{ij} F_{ij}^B \qquad (3)$$

Where $E = \begin{bmatrix} 0 & 0 & 1 \end{bmatrix}^T$ is the unit coefficient matrix, ${}^{\{B_z\}}R_{[P_{ij}]}(\alpha_{ij}) \in SO(3)$ represents the rotation matrix from the rotor frame $P_{ij}$ to the assembly frame $B_z$, which satisfied as:

$$ {}^{\{B_z\}}R_{[P_{ij}]}(\alpha_{ij}) = {}^{\{B_z\}}R_{[B_i]} \; {}^{\{B_i\}}R_{[P_{ij}]}(\alpha_{ij}) \tag{4} $$

Where ${}^{\{B_z\}}R_{[B_i]} \in SO(3)$ represents the rotation matrix from the submodule frame $B_i$ to the assembly frame $B_z$. ${}^{\{B_i\}}R_{[P_{ij}]}(\alpha_{ij}) \in SO(3)$ represents the rotation matrix from rotor frame $P_{ij}$ to submodule frame $B_i$, which satisfied as:

$$ \begin{cases} {}^{\{B_i\}}R_{[P_{ij}]}(\alpha_{i1}) = R(\sigma_1, \alpha_{i1}) \\ {}^{\{B_i\}}R_{[P_{ij}]}(\alpha_{i2}) = R(\sigma_2, \alpha_{i2}) \end{cases} \tag{5} $$

$$ R(\sigma, \alpha) = \begin{bmatrix} \cos(\sigma) & -\sin(\sigma)\cos(\alpha) & \sin(\alpha)\sin(\sigma) \\ \sin(\sigma) & \cos(\sigma)\cos(\alpha) & -\sin(\alpha)\cos(\sigma) \\ 0 & \sin(\alpha) & \cos(\alpha) \end{bmatrix} \tag{6} $$

Where $\sigma$ is the angle between the arm axis and the X-axis. According to the structure of the transverse twin-rotor UAV, it can be seen that $\sigma_1 = -\pi/2, \sigma_2 = \pi/2$.

In the assembly frame $B_z$, the rotor torque $\tau_a$ of MRBicopter is shown as follows.

$$ \tau_a = \sum_{ij} {}^{\{B_z\}}p_{[P_{ij}]} \times F_{ij}^B \tag{7} $$

Due to the action of air resistance, the yaw moment $Q$ generated by the rotor propeller is shown as follows.

$$ Q_{ij} = (-1)^{j-1} C_t \varpi_{ij} E $$
$$ Q = \sum_{ij} {}^{\{B_z\}}R_{[P_{ij}]}(\alpha_{ij}) Q_{ij} \tag{8} $$

Finally, the MRBicopter's body torque $\tau$ can be written as:

$$ \tau = \tau_a + Q \tag{9} $$

The dynamics equation of MRBicopter is established by using Newton-Euler equation.

$$ \tau = J_S\dot{\Omega} + \Omega \times J_S\Omega $$
$$ \sum_i m_i {}^{\{W_E\}}R_{[B_z]}\dot{V}_W = {}^{\{W_E\}}R_{[B_z]}F_B - \sum_i m_i gE \tag{10} $$

Where $m_i$ is the mass of the submodule and $J_S$ is the total inertia matrix of the assembly. At the same time, a kinematic

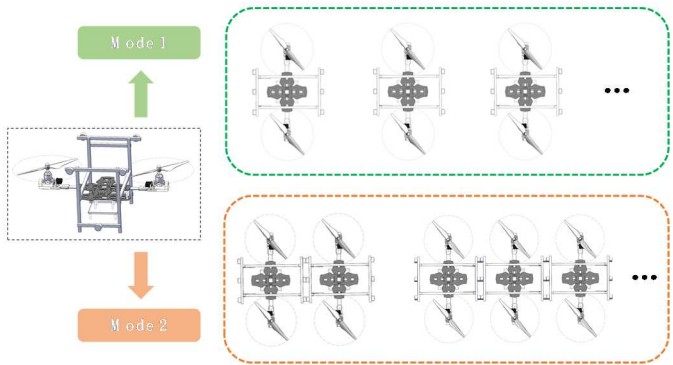

Fig.3: MRBicopter submodule(mode 1) and assembly(mode 2).

model is established on this basis, in which the position kinematic equation is expressed as:

$$ \dot{P}_W = V_W \tag{11} $$

The attitude kinematics equation is expressed as:

$$ \dot{\Theta} = W_R \cdot \Omega \tag{12} $$

## IV. CONTROL

Section IV introduces the controller design of MRBicopter single module and assembly(Fig.3), and introduces the control distribution mode of two flight modes and the feedforward Angle design of assembly [18].

### A. Controller design

Fig.5 shows the structural block diagram of the MRBicopter controller. The architecture of the controller is based on the cascade double closed-loop PID control law, with the position controller as the outer ring and the attitude controller as the inner ring. As shown in Fig.4(a), the MRBicopter submodule (mode 1) in-flight control system is an underactuated system, so we adopt the controller architecture similar to the traditional bicopter[19]. The MRBicopter assembly (mode 2) control system is an overdrive system that can achieve hovering flight at any pitch angle(Fig.4(b)).

The flight controller can be divided into four channels and output four control quantities $T_1, T_2, T_3, T_4$, which can not only control the linear displacement and angular motion of the UAV dynamics model, but also be used for decoupling the linear displacement and angular motion. The controller takes the expected position $P_{des} = \begin{bmatrix} X & Y & Z \end{bmatrix}^T$ and the expected yaw angle $\psi$ as the target control inputs respectively. $K_P^P, K_I^P, K_D^P$ is the proportion coefficient, differential coefficient and integral coefficient of the position ring respectively. Where the position controller meets:

$$ \ddot{X} = K_P^P(P - P_{des}) + K_I^P \int_0^t (P - P_{des}) + K_D^P \frac{d(\dot{P} - \dot{P}_{des})}{dt} \tag{13} $$

The attitude controller takes the expected attitude angle $\Theta_{des} = \begin{bmatrix} \phi & \theta & \psi \end{bmatrix}^T$ as input and the control quantity $T = \begin{bmatrix} T_2 & T_3 & T_4 \end{bmatrix}^T$ as output, $K_P^\Theta, K_I^\Theta, K_D^\Theta$ are the proportion

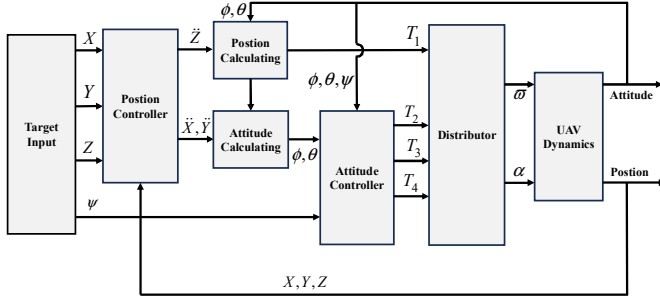

(a): Mode 1: submodule controller.

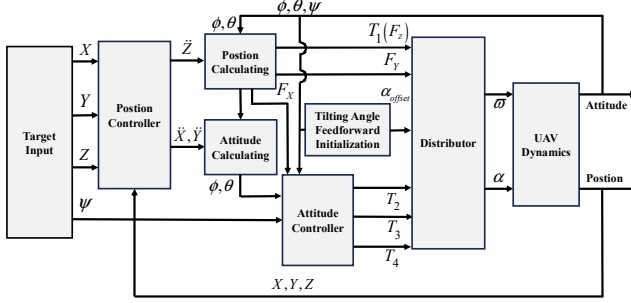

(b): Mode 2: assembly controller.

Fig.4: MRBicopter structural block diagram of flight controller.

coefficient, differential coefficient and integral coefficient of the attitude ring respectively, which are satisfied as follows:

$$T = K_P^\Theta \left( \Theta - \Theta_{des} \right) + K_I^\Theta \int_0^t \left( \Theta - \Theta_{des} \right) + K_D^\Theta \frac{d \left( \dot{\Theta} - \dot{\Theta}_{des} \right)}{dt} \quad (14)$$

### B. Tilt angle feedforward initialization calculate

The main function of feedforward initial value calculation is to solve the approximate value of rotor tilt angle when MRBicopter assembly is hovering at any pitch angle, which can effectively reduce the overshoot and response time of position control. Here, it is assumed that all rotor propellers have the same lift when the assembly hovers at any pitch angle, the hover angle is $\theta$, and the initial feedforward value of the tilt angle is $\alpha_{offset}$. As shown in Fig.5, we can establish the following force balance equation:

$$\sum_i m_i g \cos \theta = \sum_{ij} F_{ij}^B \cos \left( \alpha_{offset}^{ij} \right)$$
$$\sum_i m_i g \sin \theta = \sum_{ij} F_{ij}^B \sin \left( \alpha_{offset}^{ij} \right) \quad (15)$$

Since the resultant force in the x and y directions is zero, when the MRBicopter hovers, the initial feedforward value of the tilt angle can be obtained as:

$$\alpha_{offset}^{ij} = \theta \quad (16)$$

### C. Control distribution

The control distribution module can assign the throttle speed of the rotor and the tilt angle of the rotor in real time according

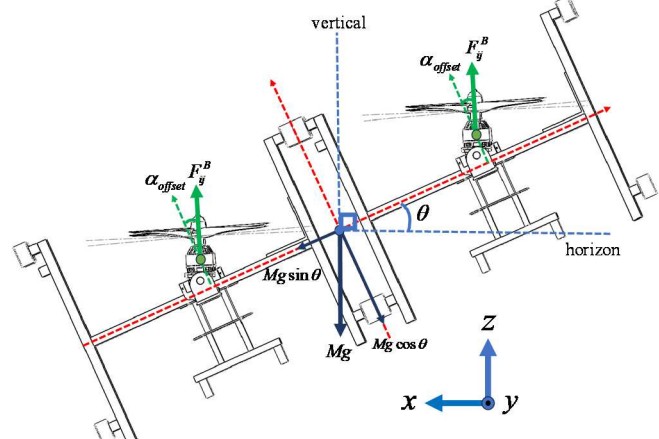

Fig.5: MRBicopter hover force analysis diagram with pitch angle.

to the mode and flight condition of the UAV, so as to achieve the purpose of controlling the attitude of the UAV.

1) Submodule control distribution

The MRBicopter submodule can be regarded as a transverse twin-rotor bicopter, with the rotor tilt axis located in the same straight line and the rotors symmetrical. Literature [20] proposed a cross-row dual-rotor UAV control method, so the control distribution mode can be transferred to the MRBicopter submodule, and the rotational speed of the left and right rotors can be expressed as:

$$\varpi_L = \sqrt{\frac{T_1}{2K_T} + T_2}$$
$$\varpi_R = \sqrt{\frac{T_1}{2K_T} - T_2} \quad (17)$$

The tilt angle of the left and right rotors can be expressed as:

$$\alpha_L = C_1 T_3 + C_2 T_4$$
$$\alpha_R = C_1 T_3 - C_2 T_4 \quad (18)$$

In equation, $C_1, C_2$ are constants.

2) Assembly control distribution

Taking MRBicopter assembly mass center $B_z$ as the center, $X_W, Y_W$ can be used to divide the rotor into four parts (Fig.6): top left rotor: $P_k \left( k = 1, 2, \cdots, n \right)$ ; Lower left rotor: $P_k \left( k = n+1, \cdots, 2n \right)$ ;top right rotor: $P_k \left( k = 2n+1, \cdots, 3n \right)$ ; lower right rotor: $P_k \left( k = 3n+1, \cdots, 4n \right)$.

Literature[20] proposes a mechanism for connecting two twin rotor modules, each of which combines two of the four propellers into a group, similar to the MRBicopter assembly structure. Therefore, the control distribution mode can be extended here. The rotor speed control distribution of the four parts can be write as follows:

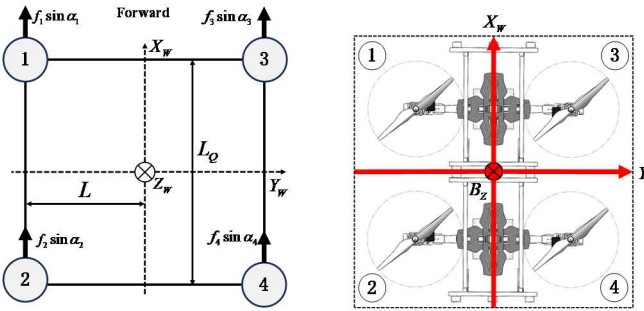

Fig.6: Mechanism model of MRBicopte.

$$\varpi_i^1 = \sqrt{\frac{F_Z}{4nK_T} + T_3 + T_2} \ (i=1,\cdots,n)$$

$$\varpi_i^2 = \sqrt{\frac{F_Z}{4nK_T} - T_3 + T_2} \ (i=n+1,\cdots,2n)$$

$$\varpi_i^3 = \sqrt{\frac{F_Z}{4nK_T} + T_3 - T_2} \ (i=2n+1,\cdots,3n)$$

$$\varpi_i^4 = \sqrt{\frac{F_Z}{4nK_T} - T_3 - T_2} \ (i=3n+1,\cdots,4n)$$

$(19)$

The MRBicopte assembly uses the $X_w$ axis to divide the left and right rotor tilt angles using different control distributions:

$$\alpha_i^1 = \alpha_{offset} + C_1 T_4 + C_2 \frac{F_Y}{4n} (i=1,2,\cdots,2n)$$

$$\alpha_i^2 = \alpha_{offset} - C_1 T_4 + C_2 \frac{F_Y}{4n} (i=2n+1,\cdots,4n)$$

$(20)$

In equation, $C_1, C_2$ are constants.

SIMULATION&EXPERIMENT

Section V mainly introduces MRBicopte submodule and assembly simulation and ground test. In order to make the simulation more realistic, we introduce the ambient wind interference model here, which can verify the robustness of the MRBicopter controller in the face of ambient wind interference.

*A. Environmental wind model*

In order to simulate the mathematical model of the atmospheric wind field as much as possible, we divide the environmental wind into constant wind, gust wind, gradient wind and random wind four parts.

Constant wind: The wind power of constant wind is a constant value $\delta$ ,the wind speed does not change. Its mathematical model of wind speed is expressed as follows:

$$V_{f1} = \delta \tag{21}$$

Gust wind: Gust wind is a kind of periodic change of wind speed in atmospheric motion, which is characterized by the sudden increase of wind speed at a certain moment and the self-weakening after a period of time. Its mathematical model can be expressed as a piecewise function:

$$V_{f2} = \begin{cases} 0 & (x<0) \\ \frac{V_m}{2}(1-\cos(\frac{\pi x}{d_m})) & (0 \le x \le d_m) \\ V_m & (x>d_m) \end{cases} \tag{22}$$

Where $V_m$ is the gust amplitude, $d_m$ is the gust length, x is the gust travel distance.

Gradient wind: Gradient wind refers to the ambient wind whose wind speed increases from zero to a certain value over time. Its mathematical model expression is as follows:

$$V_{f3} = \frac{t-t_1}{t_2-t_1} V_{f\_max} \tag{23}$$

Where $V_{f\_max}$ represents the peak of the gradual wind speed, $t_1, t_2$ represent the beginning and end of the gradual wind, respectively.

Random wind: Random wind refers to the air disturbance generated by random changes in the atmosphere. Here, we use random number generator to build a mathematical model of random wind:

$$V_{f4} = V_{f4\_max} \lambda(-10 \sim 10)\cos(\alpha t + \beta) \tag{24}$$

Where $V_{f4\_max}$ represents the theoretical peak of random wind; It is a random number generated by a random number generator, and its range is -10~10. $\alpha$ represents the average frequency of random wind speed fluctuation, with the value ranging 0.5~2rad/s. $\beta$ indicates the offset of random wind speed. The value ranges from $0.1\pi r \sim 2\pi r$ .

Therefore, if the total wind speed of the ambient wind field is $V_F$ , it can be obtained as:

$$V_F = V_{f1} + V_{f2} + V_{f3} + V_{f4} \tag{25}$$

In order to simplify the calculation, the wind speed direction is taken as the opposite of the MRBicopter's flight direction, so the air resistance generated by ambient wind field interference can be calculated:

$$F_w = \frac{1}{2} C\rho S (V_F + v_{UAV})^2 \tag{26}$$

Where $C$ represents the air resistance coefficient, the value is 0.31; $\rho$ indicates the air density, which is 1.29kg/m³. $S$ represents the windward area of MRBicopte, which is 31cm³. $v_{UAV}$ stands for flight speed.

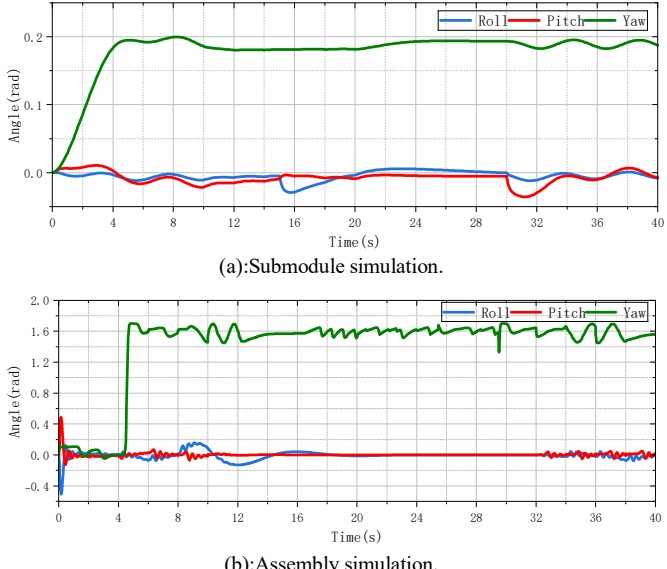

(a):Submodule simulation.

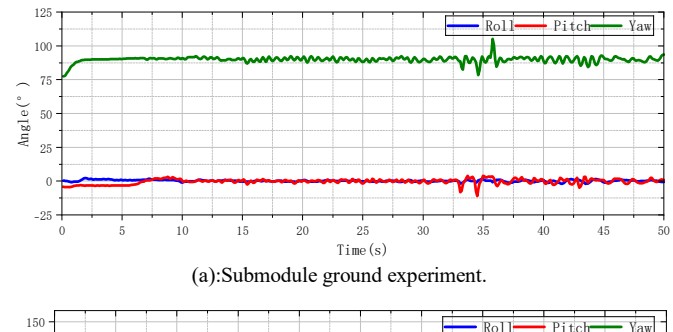

(a):Submodule ground experiment.

(b):Assembly simulation.

(b):Assembly ground experiment.

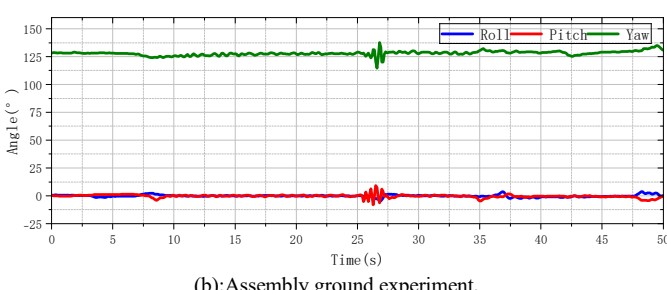

Fig.7: Simulation of MRBicopter hover under ambient wind interference, (a) MRBicopter single module; (b) MRBicopter assembly.

Fig.8: MRBicopter ground experiment under ambient wind interference.

## B. Simulation

Fig.7 shows the simulation diagram of the three-axis attitude angle of the two MRBicopter structures in hovering state under the presence of ambient wind interference. The blue line represents the roll angle tracking curve, the red line represents the pitch angle tracking curve, and the green line represents the yaw angle tracking curve.

### 1) Submodule

This experiment is a hover simulation experiment of MRBicopter submodule in the presence of ambient wind interference. The average ambient wind speed is set at 10.5m/s. The simulation experiment results are shown in Fig.7(a): the hover attitude angle oscillation of a single module does not exceed 0.05rad, which meets the design requirements.

### 2) Assembly

This experiment is a hover simulation experiment of the MRBicopter assembly in the presence of ambient wind interference. The simulation results are shown in Fig.7(b): instantaneous oscillation of >0.4rad occurs in the pitch and roll angle of the assembly at 0.3s, and the adjustment is completed within 0.2s, and the subsequent oscillation amplitude does not exceed 0.1rad. It shows that the combination controller has a strong ability to suppress the environmental wind interference.

## C. Ground experiment

In order to ensure the safety of the test, the experiment was carried out on the indoor aircraft test platform, and 1/6HP650 pneumatic industrial fan was used as the ambient wind source. The MRBicopter flight control module uses STM32F427VIT6 as the main processor; The power supply adopts LiPo(4S1P:14.8V,3000mAh); The combination docking module uses ZigBee serial communication to receive the control signal and convert it into analog PWM signal to control the on-off of the relay. The 2.4GHz14 channel communication module is used for signal sending and receiving. The experimental results are shown in Fig.8.

### 1) Submodule experiment

Two MRBicopter submodules are built here, and one of them is selected for experiment. The experimental results are shown in Fig.8(a): when there is wind interference,the average oscillation amplitude of pitch angle and roll angle of the submodule is±4.98° and the average oscillation amplitude of yaw Angle is ±7.91°, which meets the stability requirements.

### 2) Assembly experiment

The MRBicopter assembly is composed of two submodules. The experimental results are shown in Fig.8(b): when there is wind interference, the average oscillation amplitude of pitch and roll angle of the assembly is ± 5.12°, and the average oscillation amplitude of yaw angle is±7.33°, which meets the stability requirements.

## CONCLUSION

In this paper, a modular and reconfigurable multi-UAV platform MRBicopter is proposed, in which the transverse twin rotor submodule can realize structural reconstruction through the electromagnet combination docking structure, and can realize different flight states by changing the motor speed and tilt angle to meet the needs of different tasks. In order to further improve the controllability of MRBicopter and expand its application fields, improvements will be made in the following aspects in the future:

1) The fuzzy PID control algorithm is proposed to further improve the interference compensation capability of MRBicopter and improve the stability of the flight process of the assembly.

2) Structurally, further mount relevant computing units on the UAV, such as Lidar, airborne computer NUC, etc., to expand the application scenario of the MRBicopter.

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
