# OpenReview forum: "MRBicopter：Modular Reconfigurable Transverse Tilt-rotor Bicopter System"
_IEEE.org/ICIST/2024/Conference — IEEE ICIST 2024 Conference Submission_

### Official Review · Reviewer_kgFq · 2024-08-26
**Accept**

**Rating:** 10
**Confidence:** 5

**Review:**

This paper introduces a modular UAV that can realize structural combination reconstruction. Each module contains a rotor tilting structure and an active docking mechanism. Overall, the language and organization are satisfactory. It can be accepted as a conference paper.

---

### Official Review · Reviewer_8DnL · 2024-08-29
**Accept after modification**

**Rating:** 8
**Confidence:** 3

**Review:**

1、It is recommended that the authors optimize the title of each chapter.
2、In the introduction, the literature is cited in a rather confusing order.
3、Extra blank lines appear in the second contribution point.
4、For the introduction of characters appearing in the formulas of the article, there is no need to start a separate paragraph.
5、It is recommended that the authors keep the simulation times in Fig. 7 and the experimental times in Fig. 8 consistent.

---

### Official Review · Reviewer_sxF4 · 2024-08-30
**This paper can be accepted.**

**Rating:** 7
**Confidence:** 3

**Review:**

Innovation: A modular drone MRBicopter that can be combined and reconfigured is designed. Through the rotor vector tilt structure and the active combination docking mechanism, the structure can be reconfigured to adapt to different task requirements, which has certain innovation.
Advantages:
1. The experimental verification is sufficient. The environmental wind interference model is introduced, and the simulation and ground experiments are conducted to verify the robustness of the MRBicopter controller in the face of environmental wind interference.
2. The cascade double closed-loop PID control law is adopted to design the controller, which realizes the precise control of the position and attitude of the UAV.
Cons:
1. Description of docking mechanism: electromagnetic combination docking mechanism is a key part of the article, but the current description of its working principle and control response process may not be detailed enough. Elaborating on this section will give readers a better understanding of how the agency works and how it plays a role in the combination and reconfiguration of drones.
2. The test of performance indicators of MRBicopter is not comprehensive enough. Besides stability, the test and analysis of performance indicators such as endurance time, load capacity and flight speed can also be increased to evaluate the performance of UAV in a more comprehensive way.

---

### Decision · Program_Chairs · 2024-09-06

Accept (Oral)